# One-Year Visual and Refractive Outcomes of SmartPulse^®^ Technology in Transepithelial Photorefractive Keratectomy for Myopic and Astigmatic Patients

**DOI:** 10.3390/jcm13206182

**Published:** 2024-10-17

**Authors:** Daiana-Andreea Margarit, Horia Tudor Stanca, Valeria Mocanu, Mihnea Munteanu, Francis Ferrari, Suta Marius

**Affiliations:** 1Ophthalmology Department, “Victor Babes” University of Medicine and Pharmacy, 300041 Timisoara, Romania; daiana.margarit@umft.ro (D.-A.M.); mocanu.valeria@umft.ro (V.M.); suta.marius@umft.ro (S.M.); 2Ophthalmology Department, “Carol Davila” University of Medicine and Pharmacy, 050474 Bucharest, Romania; horia.stanca@umfcd.ro; 3Oftalmo Sensory-Tumor Research Center-ORL (EYEENT), 300041 Timisoara, Romania; 4Clinique Espace Nouvelle Vision, 75006 Paris, France

**Keywords:** contactless surgery, SmartSurf^ACE^, transepithelial PRK, SmartPulse^®^ technology, myopia correction, visual and refractive outcomes

## Abstract

**Background/Objectives:** This study aims to evaluate the efficacy, safety, and predictability of Transepithelial Photorefractive Keratectomy (TPRK) using the SmartPulse^®^ technology excimer laser for the correction of myopia and myopic astigmatism, assessing visual and refractive outcomes over a one-year follow-up period. **Methods:** This retrospective cohort study analyzed data from patients who underwent TPRK at the Ophthalmology Department—“Victor Babes” University of Medicine and Pharmacy in Timisoara (Romania), between January 2019 and June 2023. The procedure was performed using the SmartPulse^®^ Technology of the SmartSurf^ACE^ AMARIS 750S excimer laser (SCHWIND eye-tech-solutions, Kleinostheim, Germany). Preoperative assessments included visual acuity, refraction, and corneal measurements, with postoperative evaluations conducted for up to 12 months. **Results:** This study included 92 eyes from 46 patients (mean age 29.02 years, 63% male). At 12 months post-op, 100% achieved UDVA 20/25 or better, with an efficacy index of 1.01. Refractive accuracy was 96% within ±0.50 D of the target and astigmatism ≤ 0.50 D in 99% of eyes. The safety index was 1.01. Corneal haze occurred in 8.70% of eyes and was effectively managed with dexamethasone drops. **Conclusions:** TPRK with the SmartPulse^®^ technology excimer laser demonstrated high efficacy and safety in correcting myopia and myopic astigmatism, achieving stable visual outcomes over one year. The procedure also showed excellent predictability with a low incidence of complications, supporting its use as a reliable refractive surgery option.

## 1. Introduction

Refractive surgery has evolved significantly over the past few decades, becoming a key solution for correcting various refractive errors, such as myopia and astigmatism [1]. These procedures aim to reduce or eliminate the need for glasses or contact lenses by reshaping the cornea to improve the focus of light on the retina. Laser-assisted procedures, including photorefractive keratectomy (PRK) and laser in situ keratomileusis (LASIK), are widely used due to their high efficacy, predictability, and safety [2,3]. LASIK, in particular, gained popularity because of its rapid recovery and minimal postoperative discomfort compared to PRK. However, despite the popularity of LASIK, postoperative complications such as dry eye syndrome and flap-related issues have prompted surgeons to explore alternatives that offer similar efficacy with reduced risks in specific patient populations, such as those with thin corneas or high refractive errors [4,5]. Concerns about potential complications, such as ectasia, have led to a resurgence of interest in surface ablation techniques like PRK, especially for patients with thin corneas or those at higher risk for trauma [2]. PRK, unlike LASIK, does not involve creating a corneal flap, making it a safer option for certain patients. Moreover, PRK has demonstrated long-term stability and minimal risk of biomechanical weakening of the cornea, making it a reliable choice for patients involved in contact sports or those prone to trauma [6,7].

Transepithelial photorefractive keratectomy (TPRK) is a notable advancement in the field of refractive surgery, offering a “no-touch” approach that eliminates the need for mechanical or chemical epithelial removal. Introduced in the late 1990s, TPRK initially involved a two-step process where the epithelium was removed by phototherapeutic keratectomy (PTK) before stromal ablation. This method aimed to reduce the complications associated with manual epithelial removal, such as delayed healing and stromal haze [3,8,9,10,11]. Manual epithelial debridement has been associated with postoperative discomfort, longer healing times, and an increased risk of infection, which made the development of transepithelial techniques highly desirable in reducing these risks [7,12]. The evolution of TPRK led to the development of a more advanced single-step procedure, which has garnered attention for its efficacy and safety. In this technique, the epithelium and stroma are ablated simultaneously using an excimer laser, which delivers precise energy levels tailored to the thickness of the corneal layers. This method not only simplifies the procedure but also minimizes the risk of dehydration and enhances the accuracy of the ablation profile [13,14].

A significant innovation within this domain is the integration of SmartPulse^®^ technology, which further refines the ablation process by enhancing the smoothness of the corneal surface. SmartPulse^®^ is based on a geometric model that optimizes the laser beam’s interaction with the cornea, resulting in a smoother stromal bed post-ablation. This smoother surface facilitates quicker epithelial recovery and enhances visual clarity in the early postoperative period, which is a common concern in conventional PRK techniques [15,16,17]. This technology has been associated with faster visual recovery, reduced postoperative pain, and a lower incidence of haze compared to traditional PRK techniques [2,11,13,15,16,18]. SmartSurf^ACE^ treatment, a combination of TPRK and SmartPulse^®^ technology, represents a cutting-edge approach in refractive surgery. By utilizing this single-step, no-touch procedure, surgeons can achieve highly precise outcomes with improved patient comfort and quicker recovery times. This technique is particularly advantageous for patients with myopia and astigmatism, as it allows for the correction of these refractive errors with minimal disruption to the corneal surface [2,6].

The objective of this study was to assess the visual and refractive outcomes of TPRK for correcting myopia and myopic astigmatism using the Smart Pulse^®^ technology excimer laser. We aimed to evaluate the procedure’s efficacy, safety, predictability, and refractive stability over one year.

## 2. Materials and Methods

### 2.1. Design

This retrospective cohort study was conducted at the Ophthalmology Department of “Victor Babes” University of Medicine and Pharmacy in Timisoara (Romania), spanning from January 2019 to June 2023. The research adhered to the tenets of the Declaration of Helsinki. Due to the retrospective nature of this study, which involved analyzing pre-existing clinical data without requiring new data collection, direct patient contact, or interventions, ethical review and approval were waived. This decision was based on the fact that this study presented no risk to patient welfare as it utilized de-identified and anonymized data.

In line with ethical research practices, informed consent was obtained digitally from all participants during their initial clinical visits. This consent process clearly outlined the aims of this study, the procedures involved, and the academic use of the collected data. The consent explicitly permitted the use of participants’ data in this retrospective analysis, ensuring transparency and adherence to ethical standards.

### 2.2. Patients

This study included data from 46 patients who underwent TPRK. A comprehensive dataset was collected, encompassing demographic information, visual acuity measurements, refraction data, keratometry, recorded complications, and detailed surgery metrics. Inclusion and exclusion criteria were presented in Table 1.

### 2.3. Preoperative Examination

Before undergoing TPRK, all patients were subjected to a detailed preoperative assessment to ensure optimal surgical outcomes and patient safety.

Optometric Examination: (1) Subjective refraction was performed to determine the distance refraction. This was supplemented by cycloplegic refraction measured using an autorefractometer KR88-00 (Topcon, Tokyo, Japan), and further refined through the subjective method with the Jackson cross-cylinder to finalize the cylindrical power and axis. (2) The cover test was conducted to evaluate ocular alignment using a translucent occluder (Optometric Promotion, Burgos, Spain). (3) The near point of convergence was accurately measured to assess the patient’s ability to maintain binocular single vision at close distances.

Ophthalmological Examination: (1) Corneal topography and keratometry were measured with the Sirius Topographer (CSO, Florence, Italy). (2) Intraocular pressure and corneal biomechanics were assessed using the Topcon Tonometer (Topcon Medical Systems, Tokyo, Japan). (3) Optical coherence tomography (OCT) was utilized to evaluate epithelial thickness and retinal health Clarus OCT (Carl Zeiss Meditec, Jena, Germany). (4) Anterior segment examination was performed using a slit lamp biomicroscope (LH-2000, Indo, Barcelona, Spain). (5) Fundoscopy was conducted after cycloplegic dilation using the Binocular Indirect Ophthalmoscope (BIO) (Heine, Herrsching, Germany) to thoroughly assess the internal ocular structures. Detailed patient demographics, including age and preoperative examination data, are provided in a Appendix A available online for further reference (Appendix A).

### 2.4. Surgical Technique

The TPRK was performed utilizing the Amaris 750S laser platform (SCHWIND eye-tech-solutions GmbH, Kleinostheim, Germany) operating at a frequency of 750 kHz. The optic zone was set at a mean of 6.75 ± 0.26 mm (from 6.30 to 7.30). Energy fluence was set for high energy in 4.71 ± 0.30 mJ/cm^2^ (from 4.45 to 5.10) and low energy at 4.13 ± 55.20 mJ/cm^2^ (from 3.65 to 4.85). The high wavelength was set at 655.62 ± 22.85 nm (from 603.00 to 693.00) and the low wavelength was set at 559.82 ± 21.90 nm (from 519.00 to 589.00). Treatments were carried out with a plano target using aspheric, non-wavefront-guided profiles. Procedures commenced only when the anticipated residual stromal thickness was confirmed to exceed 300 μm.

Prior to the procedure, all patients underwent a standardized preoperative regimen. Moxifloxacin 0.5% eye drops (Vigamox, Alcon Laboratories, Fort Worth, TX, USA) were administered three times a day starting one day before surgery to reduce the risk of infection. Additionally, topical anesthetic drops, Novesine 0.4% (OmniVision, Puchheim, Germany), were applied three times to ensure patient comfort during the procedure. Immediately before surgery, the periocular area was thoroughly cleaned using a 5% povidone-iodine solution (Betadine, Avrio Health L.P., Stamford, CT, USA) to maintain a sterile surgical field and minimize the risk of contamination. A sponge saturated with balanced salt solution (BSS) was used to cleanse the corneal surface uniformly, preventing any irregularities during laser application. The laser system’s software determined the optimal ablation profile, adapting to each patient’s preoperative refractive error and corneal topography.

Central epithelial ablation was standardized at a depth of 55 μm, increasing to 65 μm at a 4 mm diameter to accommodate the paracentral cornea. Mitomycin C at 0.02% concentration (Kyowa-hakko Co., Ltd., Tokyo, Japan) was applied to the stromal surface for 30 s using a moistened sponge to mitigate postoperative haze by inhibiting fibroblast proliferation. The treated surface was then irrigated with BSS and carefully dried.

### 2.5. Postoperative Examination

In the operating room: Immediately following the procedure, patients received several topical medications to prevent infection, manage inflammation, and promote healing. Moxifloxacin 0.5% eye drops (Vigamox, Alcon Laboratories, Fort Worth, TX, USA) were administered to reduce the risk of infection. To control postoperative inflammation, Bromfenac 0.09% eye drops (Yellox, Bausch & Lomb, Laval, QC, Canada) and Cyclopentolate 1% (OmniVision, Puchheim, Germany) were applied. Tropicamide 1% (Bausch & Lomb, Rochester, NY, USA) was also administered as part of the postoperative regimen. The eyes were lubricated with Thealoz Duo (Laboratoires Théa, Clermont-Ferrand, France) to protect the ocular surface. A Night & Day bandage contact lens (Alcon, Fort Worth, TX, USA) was placed to promote epithelial healing. Finally, Tobrex ointment (Tobramycin 0.3%, Alcon Laboratories, Fort Worth, TX, USA) was applied to provide additional protection against infection.

At home: Patients were instructed to continue a strict postoperative medication regimen at home. Moxifloxacin 0.5% eye drops were prescribed four times daily for seven days. Tropicamide 1% was continued at a frequency of four times daily for three days. Starting on the third postoperative day, Flumethasone 0.1% eye drops (Flumetol S, Santen, Osaka, Japan) were introduced, with a tapering schedule: four times daily for two weeks, three times daily for the next two weeks, twice daily for the following two weeks, and once daily for the final two weeks. Bromfenac 0.09% eye drops were continued at a dosage of two times daily for two weeks to control inflammation. Thealoz Duo was recommended four times daily for up to six months to maintain corneal hydration and comfort.

### 2.6. Statistical Analysis

The statistical analysis was performed using SPSS software version 29.0 (IBM Corporation, Armonk, NY, USA). Data analysis adhered to the guidelines of the Standard Charts for Reporting Refractive Surgery [19,20,21,22,23,24]. The normality of the sample distribution was assessed using the Shapiro–Wilk test and the results confirmed that normality was achieved.

A Student’s *t*-test was applied to assess parametric dependent variables. Statistical significance was determined with a 95% confidence interval, using a *p*-value threshold of less than 0.05.

### 2.7. Visual and Refractive Outcomes Graphical Analysis

The outcomes of refractive surgery were graphically represented using the standard methods defined by Waring et al., Reinstein et al., Dupps et al., and Stulting et al. [19,20,21,22,23,24]. Visual acuity was measured using the Snellen chart, and refractive errors were measured using both automated and manual refraction techniques. Graphs A through I refer to the refractive surgery outcomes as defined by standardized protocols, explained in the following sections:

Graph A (Uncorrected Distance Visual Acuity): Cumulative percentages of eyes achieving specified levels of UDVA were plotted at 12 months post-operation. This graph helps visualize the improvement in visual acuity without correction.

Graph B (Uncorrected vs. Corrected Visual Acuity): This bar graph compares the percentage of eyes where postoperative UDVA was the same or better than preoperative CDVA, demonstrating the efficacy of the surgical correction.

Graph C (Change in Corrected Distance Visual Acuity): Displays the net changes in CDVA lines at 12 months post-surgery, categorized by the number of lines gained or lost.

Graph D (Spherical Equivalent Refraction Attempted vs. Achieved): Scatter plot illustrating the relationship between intended and achieved spherical equivalent refraction, with a linear regression line indicating predictability.

Graph E (Spherical Equivalent Refraction Accuracy): Bar graph showing the proportion of eyes within ±0.50 D and ±1.00 D of the intended refractive target.

Graph F (Spherical Equivalent Refraction Stability): Line graph depicting changes in mean spherical equivalent over time, highlighting refractive stability from preoperative to 12 months postoperative.

Graph G (Refractive Astigmatism): Histogram of refractive astigmatism comparing preoperative and postoperative values, showing the efficacy of astigmatism correction.

Graph H (Target Induced vs. Surgically Induced Astigmatism): Scatter plot of TIA versus SIA with a regression analysis, used to assess the accuracy of astigmatism correction.

Graph I (Refractive Astigmatism Angle of Error): Bar graph showing the angle of error in refractive astigmatism correction, assessing the precision of astigmatic correction orientation.

The efficacy of the procedure was quantified through the efficacy index, which is the ratio of postoperative UDVA to preoperative CDVA, thereby indicating the extent of visual improvement achieved without corrective lenses. Safety was assessed using the safety index, calculated as the ratio of postoperative to preoperative CDVA, reflecting the degree to which visual acuity was preserved after surgery. Predictability was measured by the difference between the intended and achieved refractive outcomes, with successful predictability defined as postoperative refraction within ±0.5 diopters of the target.

## 3. Results

This study evaluated 92 eyes from 46 patients, with a mean age of 29.02 ± 6.09 years, ranging from 20 to 53 years. The distribution among genders was predominantly male (63%) and all participants were Caucasian. The central corneal thickness averaged 549.52 ± 24.58 µm before surgery, with values spanning from 506 to 622 µm. Surgical parameters indicated an average ablation depth of 98.63 ± 15.82 µm (from 71.40 to 159.0 µm and a residual stromal bed thickness of 450.91 ± 30.81 µm (from 361.00 to 519.00 µm). The mean duration of the surgical procedures was 38.89 ± 9.80 s, with individual surgery times varying between 25 and 73 s.

### 3.1. Visual Outcomes

The visual outcomes of the 92 eyes treated were highly positive. As seen in Figure 1A, all eyes (100%) targeted for plano achieved an uncorrected distance visual acuity (UDVA) of 20/25 or better twelve months postoperatively. Importantly, Figure 1B demonstrates that 97% of eyes achieved the same or better UDVA than their preoperative corrected distance visual acuity (CDVA), with 100% of eyes achieving a postoperative UDVA within one Snellen line of their preoperative CDVA. This underscores the excellent predictability and consistency of visual outcomes. No eyes experienced a loss of two or more lines of CDVA after surgery (Figure 1C), indicating a high level of safety in the procedure.

The preoperative UDVA averaged 0.61 ± 0.24 logMAR, showing a significant improvement to 0.01 ± 0.02 logMAR at the 12-month follow-up (t = 24.33, *p* < 0.01). CDVA remained stable across the treatment, with preoperative CDVA at 0.00 ± 0.00 logMAR and postoperative CDVA at 0.00 ± 0.01 logMAR (t = 0.57, *p* = 0.28), further supporting the safety and precision of the procedure. The efficacy index, comparing postoperative UDVA to preoperative CDVA, was calculated at 1.01, while the safety index, comparing postoperative CDVA to preoperative CDVA, was also 1.01, reinforcing that there were no significant adverse events or visual complications.

These improvements in visual outcomes are comprehensively depicted in the Reinstein visual and refractive outcomes graphs, which provide detailed insights into each aspect of postoperative vision performance.

### 3.2. Refractive Outcomes

The refractive outcomes of the procedure demonstrated excellent accuracy and stability. As illustrated in Figure 1D, the spherical equivalent refraction shows a very close match between attempted and achieved refractive corrections, with a minor tendency towards overcorrection, particularly in eyes requiring higher levels of correction. The accuracy of the procedure is further highlighted in Figure 1E, where 96% of treated eyes were within ±0.50 D of the target spherical equivalent, and 100% were within ±1.00 D at twelve months postoperatively.

In terms of stability, Figure 1F shows that only 9% of eyes exhibited a myopic shift of greater than 0.50 D between six to twelve months post-treatment, indicating that the refractive outcomes were stable over time with minimal fluctuations.

Astigmatism correction was also highly effective. As shown in Figure 1G, 99% of eyes had postoperative astigmatism of ≤0.50 D and 100% had astigmatism of ≤1.00 D. The relationship between target and surgically induced astigmatism is shown in Figure 1H, where the data aligns closely with the ideal correction line, confirming the accuracy of astigmatism correction. The angle of error for astigmatism correction is illustrated in Figure 1I, which shows that 88% of eyes had an astigmatic axis rotation within ±5 degrees of the intended correction, with only 5% showing a larger deviation of 5 to 15 degrees, and none showing a deviation greater than 15 degrees.

These refractive outcomes indicate that the procedure was highly effective in achieving the intended refractive targets, with excellent predictability, accuracy, and stability, and with minimal postoperative refractive shifts or astigmatic axis deviations.

### 3.3. Complications

Among the 92 eyes studied, 8 developed postoperative corneal haze, resulting in a complication rate of approximately 8.70%. This mild haze was uniformly treated to mitigate any long-term visual impairment. The treatment involved the administration of Dexamethasone 0.1% eye drops (Maxidex, Alcon Pharmaceuticals, Fort Worth, TX, USA). The prescribed regimen was to apply these drops four times daily for the first week, followed by a gradual taper over the next four to six weeks, depending on the patient’s response to treatment. This approach aims to prevent the proliferation of myofibroblasts and excessive collagen deposition, which contribute to haze formation.

**Figure 1 jcm-13-06182-f001:**
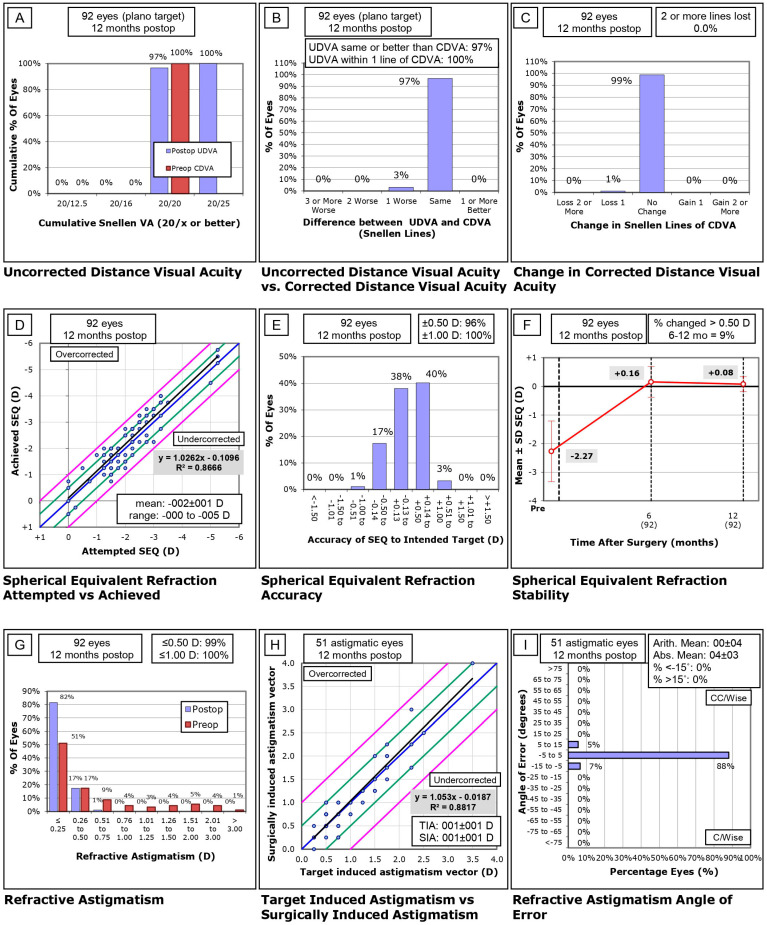
Visual and refractive outcomes following SmartPulse^®^-enhanced transepithelial PRK based on the standard nine graphs design by Reinstein. (**A**) Uncorrected distance visual acuity. (**B**) Uncorrected distance visual acuity vs. corrected distance visual acuity. (**C**) Change in corrected distance visual acuity. (**D**) Spherical equivalent refraction attempted vs. achieved. (**E**) Spherical equivalent refraction accuracy (**F**) Spherical equivalent refraction stability. (**G**) Refractive astigmatism. (**H**) Target-induced astigmatism vs. surgically-induced astigmatism (**I**) Refractive astigmatism angle of error.

### 3.4. Keratometry Evaluations

In the keratometry assessment, mean keratometry readings decreased significantly from a preoperative average of 43.70 ± 1.46 D to a postoperative average of 41.82 ± 1.61 D (t = 8.28, *p* < 0.01). The maximum keratometry decreased from 44.24 ± 1.48 D preoperatively to 42.16 ± 1.59 D postoperatively (t = 9.11, *p* < 0.01), and the minimum keratometry readings lowered from 43.18 ± 1.53 D to 41.49 ± 1.65 D (t = 7.18, *p* < 0.01). These significant changes in the corneal curvature are captured in Figure 2, depicting detailed shifts in keratometry from preoperative to postoperative stages. The axes of keratometry exhibited minimal variation postoperatively compared to preoperative measurements. Preoperatively, the mean axis for maximum keratometry (K_max_) was 92.34 ± 27.61 degrees, which slightly increased to 92.87 ± 26.30 degrees postoperatively (t = −0.29, *p* < 0.38). This minor change indicates a stable orientation of the corneal peak curvature following the procedure. In contrast, the mean axis for minimum keratometry (K_min_) showed a more notable shift, from 75.56 ± 75.38 degrees preoperatively to 82.68 ± 76.23 degrees postoperatively, but no significance was founded (t = −0.88, *p* < 0.18).

## 4. Discussion

Our study demonstrated that TPRK using SmartPulse^®^ Technology provides significant improvements in visual outcomes, resulting in excellent UDVA and stable refractive outcomes over a one-year follow-up period. The procedure showed high efficacy and safety, with minimal complications, such as manageable corneal haze. The findings suggest that TPRK with SmartPulse^®^ Technology is an effective and reliable option for correcting myopia and myopic astigmatism, achieving precise and predictable results. These outcomes align with and add to the growing body of evidence supporting the use of advanced laser technologies in refractive surgery.

Our study’s results are consistent with a variety of findings from earlier and more recent research on TPRK in terms of efficacy, safety, complications, and predictability [9,13,14,15,16,25,26,27,28,29,30]. Starting with efficacy, Aslanides et al. [15] demonstrated that TPRK with SmartPulse^®^ technology leads to excellent visual recovery in the postoperative period. This aligns with our findings, where we observed excellent visual improvement, with 100% of patients achieving a UDVA of 20/25 or better at the 12-month follow-up. This visual recovery supports the idea that integrating advanced technology, such as SmartPulse^®^, enhances the procedure’s efficacy.

In terms of safety, particularly regarding visual outcomes, Bakshk et al. [14] found that TPRK provided similar long-term visual outcomes as alcohol-assisted PRK (AAPRK) but with a better safety profile during the early postoperative period, including fewer incidences of corneal haze. Our study also observed minimal postoperative haze, with only 8.7% of eyes affected, further supporting the safety of TPRK in maintaining clear visual outcomes over time. Alasmari et al. [25] confirmed the safety and effectiveness of TPRK in treating mild myopia, noting that all patients achieved excellent postoperative UDVA with minimal complications. Our study extends these findings to a broader range of refractive errors, showing that TPRK remains effective and safe across different degrees of myopia and astigmatism, achieving high rates of visual acuity with few complications.

When considering predictability, Zhang et al. [9] found that TPRK was highly predictable for high myopia correction, with a significant proportion of patients achieving outcomes within ±0.50 D of the target refraction. Our results, showing that 96% of eyes were within ±0.50 D of the intended correction, align with Zhang et al., demonstrating that TPRK offers precise refractive outcomes. Regarding complications, Kaluzny et al. [28]. reported minimal corneal haze in their long-term study of mixed astigmatism correction using TPRK. Similarly, our study observed a low rate of mild corneal haze, which was effectively managed with topical steroids, underscoring the procedure’s safety profile in preventing significant long-term visual complications. Finally, Ho et al. [13] reported excellent predictability and safety in moderate to high myopia cases treated with TPRK, which parallels our findings in moderate myopia. Our study confirmed that TPRK, especially when enhanced with SmartPulse^®^ technology, consistently delivers safe and predictable outcomes, even in cases requiring significant refractive corrections.

### 4.1. Limitations

Despite the positive outcomes observed, this study has several limitations. First, as a retrospective cohort study, it is subject to inherent biases related to data collection and analysis, including the lack of randomization and control groups. The relatively short follow-up period of one year may not capture long-term stability and potential late-onset complications, such as regression or ectasia. Additionally, this study was conducted at a single center with a relatively small sample size, which may limit the generalizability of the findings to broader populations. A limitation of this study is that although intraocular pressure (IOP) was measured as part of the standard refractive surgery protocol, specific values were not recorded for patients with normal IOP. Furthermore, while this study focused on myopia and myopic astigmatism, it did not include patients with higher degrees of refractive error or other forms of astigmatism, which may limit the applicability of the results to these groups.

### 4.2. Future Lines of Research

Future research should explore the application of TPRK with SmartPulse^®^ Technology across different laser settings to optimize outcomes for various patient profiles. Studies could investigate the efficacy and safety of this technology in correcting higher degrees of myopia and astigmatism, as well as its effectiveness in treating hyperopia and presbyopia, expanding its use to a broader range of refractive errors. Additionally, long-term studies with follow-up periods of 5 to 10 years are needed to evaluate the durability of refractive outcomes and the potential for refractive regression over time. Research should also focus on assessing TPRK as a technique to enhance outcomes in cases of refractive surprise or changes following other refractive procedures, such as SMILE or LASIK. Finally, integrating advanced imaging and diagnostic tools, such as wavefront-guided ablation or customized treatment planning, could further refine TPRK, improving precision and patient satisfaction. In future studies, we aim to assess postoperative corneal clarity, including the incidence of corneal haze and visual disturbances such as halos and glare, to provide a more comprehensive understanding of long-term visual outcomes.

## 5. Conclusions

This study demonstrates that TPRK using SmartPulse^®^ Technology on the SmartSurf^ACE^ AMARIS excimer laser is a highly effective and safe method for correcting myopia and myopic astigmatism. Over the course of one year, patients experienced significant improvements in UDVA, with a high rate of predictability and stability in refractive outcomes. The procedure maintained a strong safety profile, with no significant loss of CDVA and a low incidence of complications, such as corneal haze, which were effectively managed. These findings support the use of TPRK with SmartPulse^®^ Technology as a reliable option for achieving precise and stable visual outcomes in patients with myopia and myopic astigmatism, offering a compelling alternative to other refractive surgery techniques.

## Figures and Tables

**Figure 2 jcm-13-06182-f002:**
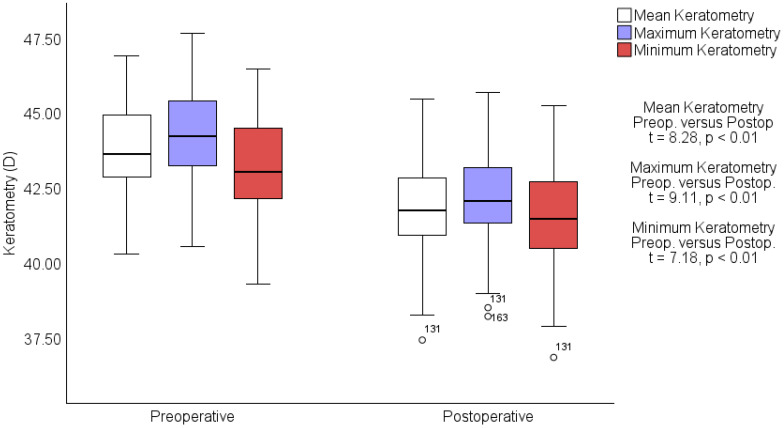
Preoperative vs. postoperative keratometry measurements. Box-and-whisker plots showing changes in mean (white), maximum (blue), and minimum (red) keratometry measurements in diopters (D) from preoperative to postoperative assessments. Statistically significant decreases are noted across all measurements with t-values and *p*-values indicating robust changes. Outliers are marked with circles.

**Table 1 jcm-13-06182-t001:** Inclusion and Exclusion Criteria on TPRK Using SmartPulse^®^ Technology.

Inclusion Criteria	Exclusion Criteria
Patients aged 18 years or older with a stable refractive error (no change of 0.50 D or more within the last year)	Systemic conditions that might affect wound healing, such as diabetes or autoimmune diseases
Recorded myopia of up to −5.50 D and astigmatism of up to −4.00 D	History of keratoconus, uncontrolled glaucoma, or herpes simplex virus eye disease
Normal corneal topography with no indications of keratoconus or other corneal ectatic disorders	Significant dry eye syndrome that could not be managed medically
Adequate corneal thickness defined as over 500 microns to ensure a postoperative residual stromal bed of at least 300 μm	Pregnancy or nursing, due to potential variability in refraction during and after pregnancy
No previous ocular surgeries or ocular pathologies affecting surgical outcomes	

## Data Availability

The raw data supporting the conclusions of this article will be made available by the authors on request.

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
