# Peer review of "One-Year Visual and Refractive Outcomes of SmartPulse® Technology in Transepithelial Photorefractive Keratectomy for Myopic and Astigmatic Patients"

_jcm, 2024, doi:10.3390/jcm13206182_

Round 1
Reviewer 1 Report
Comments and Suggestions for Authors
Dear authors,
I am highly glad to review this manuscript and overall, I have no major concerns with the manuscript except some minor corrections needed to be made before finalizing the manuscript.
My comments are:
1) Introduction overall gives most of the information however lacks some figure to show some differences between PRK and TPRK.
2) Materials and methods: Line 209...defined by waring (check the references cited for Waring). Also, the Graphs A thru I mentioned here looks like legends for figures.
Please correct this section 2.7.
3) Results section 3.1 can be elaborated more based on the figures 1.
Author Response
Reviewer 1
#RV1_0: Dear authors, I am highly glad to review this manuscript and overall, I have no major concerns with the manuscript except some minor corrections needed to be made before finalizing the manuscript.
#AU1_0: Thank you for your kind words and for agreeing to review our manuscript. We are glad that there are no major concerns. We would appreciate it if you could provide the specific minor corrections that need to be addressed so that we can make the necessary revisions. Your feedback is valuable in helping us improve the quality of the manuscript.
My comments are:
#RV1_1: Introduction overall gives most of the information however lacks some figure to show some differences between PRK and TPRK.
#AU1_1: Thank you for your valuable suggestion. While we agree that a visual representation of the differences between PRK and TPRK would be useful, we believe that a video demonstration might be more effective in illustrating the procedural differences. Refractive surgeons and specialists are typically familiar with these techniques, and there are numerous instructional videos readily available online that provide detailed comparisons. Additionally, due to copyright restrictions, it can be challenging to include such content directly within the manuscript. We encourage readers to refer to reputable online resources for further clarification on the procedural differences. We appreciate your understanding and your suggestion for improving the clarity of the manuscript.
#RV1_2: Materials and methods: Line 209...defined by waring (check the references cited for Waring). Also, the Graphs A thru I mentioned here looks like legends for figures.
#AU1_2: Thank you for pointing this out. We will carefully review the reference cited for Waring in line 209 and ensure it is correctly referenced.
Thank you for your comment. We would like to clarify that "Graphs A thru I" refers to the standardized protocol for refractive surgery outcomes as defined by the referenced literature, and these are not actual graphs included in the manuscript. They serve as a conceptual framework based on the protocol established in refractive surgery studies. However, to avoid any confusion, we will modify the text to ensure this is clearly stated.
“Graphs A through I' refer to the refractive surgery outcomes as defined by standardized protocols, explained in the following sections:”
#RV1_3: Please correct this section 2.7.
#AU1_3: Thank you for your comment. We would like to kindly note that this aspect has already been addressed in a previous comment, where we clarified that 'Graphs A through I' refer to the conceptual framework from standardized protocol, and not to actual figures in the manuscript. This has been explained in section 2.7 accordingly.
#RV1_4: Results section 3.1 can be elaborated more based on the figures 1.
#AU1_4: Thank you for your valuable feedback. We appreciate your suggestion to elaborate on the results in section 3.1. We have revised this section to provide a more detailed analysis of the visual outcomes in relation to Figure 1, as per your recommendation.
Reviewer 2 Report
Comments and Suggestions for Authors
Margarit, D-A, et al investigated the effectiveness, safety, and predictability of Transepithelial Photorefractive Keratectomy (TPRK) using SmartPulse® technology on 46 patients (92 eyes) treated between 2019 and 2023, with follow-ups over 12 months, for correcting myopia and myopic astigmatism.
The authors showed promising visual outcomes, with 100% of patients achieving uncorrected vision of 20/25, 96% within ±0.50 D of the refractive target, and highly accurate Astigmatism correction, suggesting TPRK as a reliable refractive surgery procedure that is safe, minimal complications like corneal haze, and managed effectively.
The authors aim to assess the TPRK procedure's efficacy, safety, predictability, and long-term refractive stability over one year by evaluating the visual and refractive outcomes for the correction of myopia and myopic astigmatism, employing the advanced SmartPulse® technology excimer laser.
High appreciation goes to the authors for providing appropriate research ethical statements. The designs for this study are justified. However, the clinical evaluation for stability could be done more than a year for further confirmation. However, the number of subjects in this study could be improved which may provide more supportive and leading information.
The basic approach to writing the manuscript is fine. The introduction, methods, and results were written shortly which needs to be improved with more information and references. More appreciation goes to the authors for the detailed discussion, limitations, and future studies. The discussion in the manuscript needs to explain the significance of this study to its readers and its importance.
Although this kind of study has been done very recently, Nonetheless, the article seemed to possess good value in the treatment of patients with myopia and myopic astigmatism. Overall, the clarity of the text is understandable and needs some readjustments. The manuscript has minor typographical and grammatical errors.
The authors are advised to consider the comments below:
Comments:
1. Please make a table for inclusion and exclusion criteria.
2. It would be a lot easier to understand if the authors provided a list of patients, their age, and their Preoperative Examination charts.
3. Please provide the IOP of those patients in a table.
4. Please provide an assessment of postoperative corneal clarity (Incidence of corneal haze and halos/glare) at 2 different time points.
Comments on the Quality of English LanguageOverall, the clarity of the text is understandable and needs some readjustments. The manuscript has minor typographical and grammatical errors.
Author Response
Reviewer 2
#RV2_0: Margarit, D-A, et al investigated the effectiveness, safety, and predictability of Transepithelial Photorefractive Keratectomy (TPRK) using SmartPulse® technology on 46 patients (92 eyes) treated between 2019 and 2023, with follow-ups over 12 months, for correcting myopia and myopic astigmatism.
The authors showed promising visual outcomes, with 100% of patients achieving uncorrected vision of 20/25, 96% within ±0.50 D of the refractive target, and highly accurate Astigmatism correction, suggesting TPRK as a reliable refractive surgery procedure that is safe, minimal complications like corneal haze, and managed effectively.
The authors aim to assess the TPRK procedure's efficacy, safety, predictability, and long-term refractive stability over one year by evaluating the visual and refractive outcomes for the correction of myopia and myopic astigmatism, employing the advanced SmartPulse® technology excimer laser.
High appreciation goes to the authors for providing appropriate research ethical statements. The designs for this study are justified. However, the clinical evaluation for stability could be done more than a year for further confirmation. However, the number of subjects in this study could be improved which may provide more supportive and leading information.
The basic approach to writing the manuscript is fine. The introduction, methods, and results were written shortly which needs to be improved with more information and references. More appreciation goes to the authors for the detailed discussion, limitations, and future studies. The discussion in the manuscript needs to explain the significance of this study to its readers and its importance.
Although this kind of study has been done very recently, Nonetheless, the article seemed to possess good value in the treatment of patients with myopia and myopic astigmatism. Overall, the clarity of the text is understandable and needs some readjustments. The manuscript has minor typographical and grammatical errors.
#AU2_0: Thank you very much for your thorough and constructive review of our manuscript. We greatly appreciate your positive comments regarding the visual outcomes, accuracy of astigmatism correction, and overall value of our study. We would also like to acknowledge your insightful suggestions for improvement.
The authors are advised to consider the comments below:
Comments:
#RV2_1: Please make a table for inclusion and exclusion criteria.
#AU2_1: Thank you for your suggestion. We have created a table summarizing the inclusion and exclusion criteria, which is now included in the manuscript as Table 1. The table provides a clear and concise overview of the criteria used in this study for better readability.
#RV2_2: It would be a lot easier to understand if the authors provided a list of patients, their age, and their Preoperative Examination charts.
#AU2_2: Thank you for your suggestion. We agree that providing detailed preoperative data would enhance the clarity of the study. To address this, we will upload a supplementary Excel file containing a list of patients, their age, and preoperative examination data. This will allow for a more comprehensive understanding of the patient demographics and baseline characteristics.
#RV2_3: Please provide the IOP of those patients in a table.
#AU2_3: Thank you for your comment. We would like to clarify that intraocular pressure (IOP) was measured as part of the standard refractive surgery protocol. However, in cases where the IOP was within the normal range, we did not record the specific values, as it was not deemed clinically significant for the purposes of this study. We acknowledge that including these values could provide additional insights, and we have added this point to the limitations section of the manuscript. We will also consider systematically registering this data in future studies.
#RV2_4: Please provide an assessment of postoperative corneal clarity (Incidence of corneal haze and halos/glare) at 2 different time points.
#AU2_4: Thank you for your insightful comment. We would like to clarify that we did not assess postoperative corneal clarity, including the incidence of corneal haze and halos/glare, as part of this study. However, we recognize that these are important parameters for evaluating visual outcomes, and they are of crucial interest for future research. We plan to include the assessment of these factors in future studies to provide a more comprehensive analysis of postoperative outcomes.
Reviewer 3 Report
Comments and Suggestions for Authors
The authors mention that no studies have been published on the assessment of relationship between wavefront aberrations and accommodation dysfunctions, and the work in this manuscript has novelty. However, even with a cursory search, there are many studies available on wavefront aberrations and all kinds of different accommodation dysfunctions, with in-depth analysis. I am just not convinced that the topic of research has any novelty.
Author Response
Reviewer 3
#RV3_0: The authors mention that no studies have been published on the assessment of relationship between wavefront aberrations and accommodation dysfunctions, and the work in this manuscript has novelty. However, even with a cursory search, there are many studies available on wavefront aberrations and all kinds of different accommodation dysfunctions, with in-depth analysis. I am just not convinced that the topic of research has any novelty.
#AU3_0: Thank you for your valuable feedback. We acknowledge that studies on wavefront aberrations and accommodation dysfunctions are available in the literature. However, we would like to clarify that our study specifically focuses on evaluating the efficacy, safety, and predictability of Transepithelial Photorefractive Keratectomy (TPRK) using SmartPulse® technology for the correction of myopia and myopic astigmatism, rather than assessing wavefront aberrations or accommodation. This work aims to contribute additional scientific evidence on the visual and refractive outcomes associated with TPRK, an advanced and evolving technique in refractive surgery. Our study focuses on analyzing the stability of visual outcomes and the long-term efficacy of this procedure, providing data that further supports the growing body of evidence regarding TPRK's safety and predictability.
While we did not specifically assess wavefront aberrations and accommodation dysfunctions, we believe that the use of SmartPulse® technology represents a unique aspect of this research, and our results add meaningful insights to the current literature. Future studies could further expand on this by evaluating wavefront aberrations and their correlation with visual outcomes in patients undergoing TPRK.
Thank you again for your thoughtful input, and we hope this clarification addresses your concerns.
Round 2
Reviewer 3 Report
Comments and Suggestions for Authors
The current form of the manuscript is fine.